# Plasma extracellular vesicle synaptic proteins as biomarkers of clinical progression in patients with Parkinson's disease

**Chien-Tai Hong[1,2,3], Chen-Chih Chung[1,2,3], Ruan-Ching Yu[4], Lung Chan[1,2,3]***

[1]Department of Neurology, Shuang Ho Hospital, Taipei Medical University-Shuang Ho Hospital, New Taipei City, Taiwan; [2]Department of Neurology, School of Medicine, College of Medicine Taipei Medical University-Shuang Ho Hospital, Taipei, Taiwan; [3]Taipei Neuroscience Institute, Taipei Medical University, Taipei, Taiwan; [4]Division of Psychiatry, University College London, London, United Kingdom

*For correspondence:
cjustinmd@tmu.edu.tw

**Competing interest:** The authors declare that no competing interests exist.

**Abstract** Synaptic dysfunction plays a key role in Parkinson's disease (PD), and plasma extracellular vesicle (EV) synaptic proteins are emerging as biomarkers for neurodegenerative diseases. Assessment of plasma EV synaptic proteins for their efficacy as biomarkers in PD and their relationship with disease progression was conducted. In total, 144 participants were enrolled, including 101 people with PD (PwP) and 43 healthy controls (HCs). The changes in plasma EV synaptic protein levels between baseline and 1-year follow-up did not differ significantly in both PwP and HCs. In PwP, the changes in plasma EV synaptic protein levels were significantly associated with the changes in Unified Parkinson's Disease Rating Scale (UPDRS)-II and III scores. Moreover, PwP with elevated levels (first quartile) of any one plasma EV synaptic proteins (synaptosome-associated protein 25, growth-associated protein 43 or synaptotagmin-1) had significantly greater disease progression in UPDRS-II score and the postural instability and gait disturbance subscore in UPDRS-III than did the other PwP after adjustment for age, sex, and disease duration. The promising potential of plasma EV synaptic proteins as clinical biomarkers of disease progression in PD was suggested. However, a longer follow-up period is warranted to confirm their role as prognostic biomarkers.

## eLife assessment

This **useful** study presents data regarding the presence of synaptic proteins in the extracellular vesicle pool present in the blood of Parkinson's patients and non-Parkinson neurological outpatients, trying to correlate changes in such levels with the progression of Parkinson's symptoms. The results are semi-quantitative and preliminary, suggesting that these biomarkers could be used in the follow-up of a specific group of Parkinson's patients. The evidence is **incomplete** at this point, and more quantitative approaches are required to propose this correlation. The isolation of extracellular vesicles was appropriate as revealed by their sizes, but they are not exclusively from neuronal origin. The presented approach is not ready to be used in the clinical setting.

## Introduction

Parkinson's disease (PD) is the second most common neurodegenerative disease (*de Lau and Breteler, 2006*) that is well known for its progression, which involves increased disability and burden (*Bloem et al., 2021*). Worsening is noted not only in motor symptoms but also in nonmotor ones, particularly

cognition. The rate of disease progression varies among people with PD (PwP). In the ongoing Parkinson's Progression Markers Initiative cohort study, approximately one-third of untreated PwP exhibited rapid progression during the first 2 years of follow-up. By contrast, the remaining PwP had a slow progressive course (*Chen-Plotkin et al., 2018*). Unfortunately, no disease-modifying therapy for halting disease progression and no predictor for assessing disease progression are available.

Synapses are sites of neuronal communication, and synaptic degeneration is an early functional pathogenic event in neurodegenerative diseases such as Alzheimer's disease (AD) and PD (*Picconi et al., 2012*; *Arendt, 2009*). Postmortem studies have revealed a substantial loss of dopamine terminals in the putamen and amygdala (*Iseki et al., 2001*; *Kordower et al., 2013*). In addition, the loss of glutamatergic corticostriatal synapses has been reported (*Stephens et al., 2005*). Mitochondrial dysfunction contributes to some of the synaptic loss in PD, and the buildup of α-synuclein, a key synaptic protein, plays a significant role in the development of PD (*Gcwensa et al., 2021*). Assessment of synaptic proteins in the cerebrospinal fluid (CSF) can reflect synaptic loss in patients with neurological diseases and is a key area of research interest. Significant research has been directed toward evaluating synaptic changes in CSF to enhance early diagnosis of neurodegenerative diseases before neuron loss and track disease progression (*Lleó et al., 2019*; *Milà-Alomà et al., 2021*; *Bereczki et al., 2017*). Several synaptic proteins are promising biomarkers of synaptic function. Synaptosome-associated protein 25 (SNAP-25) is a presynaptic protein that plays a crucial role in neuronal survival, vesicular exocytosis, and neurite outgrowth (*Antonucci et al., 2016*). Increased CSF levels of SNAP-25 have been reported in PwP (*Bereczki et al., 2017*). In addition, growth-associated protein 43 (GAP-43) is a presynaptic protein anchored to the cytoplasmic side of the presynaptic plasma membrane (*Benowitz and Routtenberg, 1997*). CSF levels of GAP-43 have been reported to be significantly higher in patients with AD than in healthy controls (HCs) (*Sjögren et al., 2001*). Synaptotagmin-1 is a calcium sensor vesicle protein that is vital for rapid synchronous neurotransmitter release in hippocampal neurons (*Courtney et al., 2019*). Significantly increased CSF levels of synaptotagmin-1 have been reported in patients with AD and mild cognitive impairment (*Öhrfelt et al., 2019*). Regarding PD, one study reported an increase in the level of CSF SNAP-25 but not Ras-related protein 3A or neurogranin. Moreover, treated PwP exhibited higher CSF levels of SNAP-25 than did their drug-naïve counterparts (*Bereczki et al., 2017*).

However, collecting CSF is somewhat invasive and necessary for clinical assessments, but it may cause side effects like headaches after the procedure. Although studies have assessed blood biomarkers for PD diagnosis and progression, the results are conflicting. A lack of correlation between the peripheral blood content and the brain because of the blood–brain barrier (BBB) is a major obstacle to the identification of blood biomarkers for neurodegenerative diseases (*Qin et al., 2016*). Assessment of peripheral blood extracellular vesicle (EV) proteins can be an alternative approach. EVs are tiny vesicles covered with a lipid membrane. They contain proteins, lipids, and nucleic acid responsible for cell-to-cell signal transmission. The integrity of EVs can be maintained when crossing the BBB (*Jan et al., 2017*). Plasma EV biomarkers are being rapidly developed for PD (*Leggio et al., 2021*). EV-cargo α-synuclein has been the most studied target that has exhibited strong potential for distinguishing PwP from HCs and other patients with atypical parkinsonism (*Chung et al., 2021b*; *Niu et al., 2020*; *Zheng et al., 2021*; *Si et al., 2019*; *Stuendl et al., 2021*). EV-cargo tau, β-amyloid, neurofilament light chain, brain-derived neurotrophic factor, and insulin receptor substrate have also been assessed in PwP (*Athauda et al., 2019*; *Chou et al., 2020*; *Chung et al., 2020d*; *Chung et al., 2020a*; *Chung et al., 2021a*). These results indicate the potential role of EV content as biomarkers in PD.

The levels of synaptic proteins in blood exosomes, a specific type of EV, decrease in patients with AD and frontotemporal dementia (*Goetzl et al., 2016*). Moreover, blood exosomal SNAP-25, GAP-43, neurogranin, and synaptotagmin-1 levels are lower in patients with AD. A combination of exosomal synaptic protein biomarkers could predict cognitive impairment (*Jia et al., 2021*). A cross-sectional study showed that PwP have lower levels of synaptic proteins in neuron-derived exosomes in their blood compared to healthy individuals, allowing for an approximately 80% accurate distinction between PwP and HCs (*Agliardi et al., 2021*). Considering the plasma EVs remain stable up to 90 d (*Kalra et al., 2013*), this prevents the fluctuation of free-form synaptic proteins because of transient surge or degradation. Moreover, SNAP-25 is transported in the blood by EVs (*Agliardi et al., 2019*). However, there is no information from the cohort study of PD. Therefore, the association between

**Table 1.** Demographic data of study participants.

|  | HCs (*n* = 43) | PwP (*n* = 101) |
|---|---|---|
| Age (y) | 65 (10.24) | 69 (7.76) |
| Women | 15 | 48 |
| Baseline |  |  |
| MMSE | 27 (3.92) | 26 (4.15) |
| MoCA | 23 (4.63) | 21 (5.70) |
| Disease duration (y) | - | 2 (2.24) |
| UPDRS-II | - | 8 (5.58) |
| UPDRS-III | - | 22 (9.30) |
| 1-year follow-up |  |  |
| MMSE | 28 (4.12) | 27 (5.61) |
| MoCA | 24 (5.82) | 23 (6.45) |
| UPDRS-II | - | 11 (6.41) |
| UPDRS-III | - | 19 (9.39) |

Data is presented as median (standard deviation).

HC = healthy control; PwP = people with Parkinson's disease; MMSE = Mini-Mental State Examination; MoCA = Montreal Cognitive Assessment; UPDRS = Unified Parkinson's Disease Rating Scale.

plasma EV synaptic proteins and PD progression was assessed and determined whether plasma EV synaptic proteins could be used as clinical biomarkers to predict the progression of PD.

## Results

The participants' demographic data at baseline and 1-year follow-up are presented in *Table 1*. In total, 144 participants (101 PwP and 43 HCs, all of them are Taiwanese) were followed up. No significant difference was noted in plasma EV SNAP-25, GAP-43, and synaptotagmin-1 levels at baseline and follow-up between PwP and HCs after adjustment for age and sex (*Figure 1A* [representative image] and *Figure 1B–D* [dot plot]).

The association between changes in plasma EV synaptic protein levels and clinical parameters in PwP was assessed through a generalized linear model (*Table 2*). The changes in the total score of Unified Parkinson's Disease Rating Scale (UPDRS)-II was positively associated with the change in plasma EV synaptic proteins (SNAP-25, GAP-43, and synaptotagmin-1). The changes in total score of UPDRS-III and akinetic rigidity (AR) subscore were significantly associated with the changes in plasma EV GAP-43 and synaptotagmin-1. The changes in Mini-Mental Status Examination (MMSE) and Montreal Cognitive Assessment (MoCA) scores were nonsignificantly associated with the changes in plasma EV synaptic protein levels.

The association between the severity of clinical parameters of PD at follow-up and baseline plasma EV synaptic protein levels was further evaluated. After adjustment for age, sex, and disease duration, the plasma EV SNAP-25, GAP-43, and synaptotagmin-1 levels were nonsignificantly associated with the UPDRS-II, UPDRS-III, MMSE, and MoCA scores at follow-up (*Figure 2*; for details, refer to *Supplementary file 1*). However, when UPDRS-III scores were divided into tremor, AR, and postural instability and gait disturbance (PIGD) subscores, the initial levels of plasma EV SNAP-25 and GAP-43 showed a significant positive correlation with PIGD subscores at follow-up. Additionally, a similar trend was observed with the plasma EV synaptotagmin-1 level.

Participants with PD were grouped based on their baseline plasma EV synaptic protein levels using the first quartile as a cutoff. Overall, PwP with elevated baseline levels of plasma EV synaptic proteins had poor total scores of UPDRS-II and UPDRS-III and poor PIGD subscores of UPDRS-III (*Table 3*). In contrast, a significant greater improvement in the tremor score was noted in the PwP with elevated baseline levels of plasma EV synaptic proteins. Moreover, PwP with elevated baseline levels of plasma

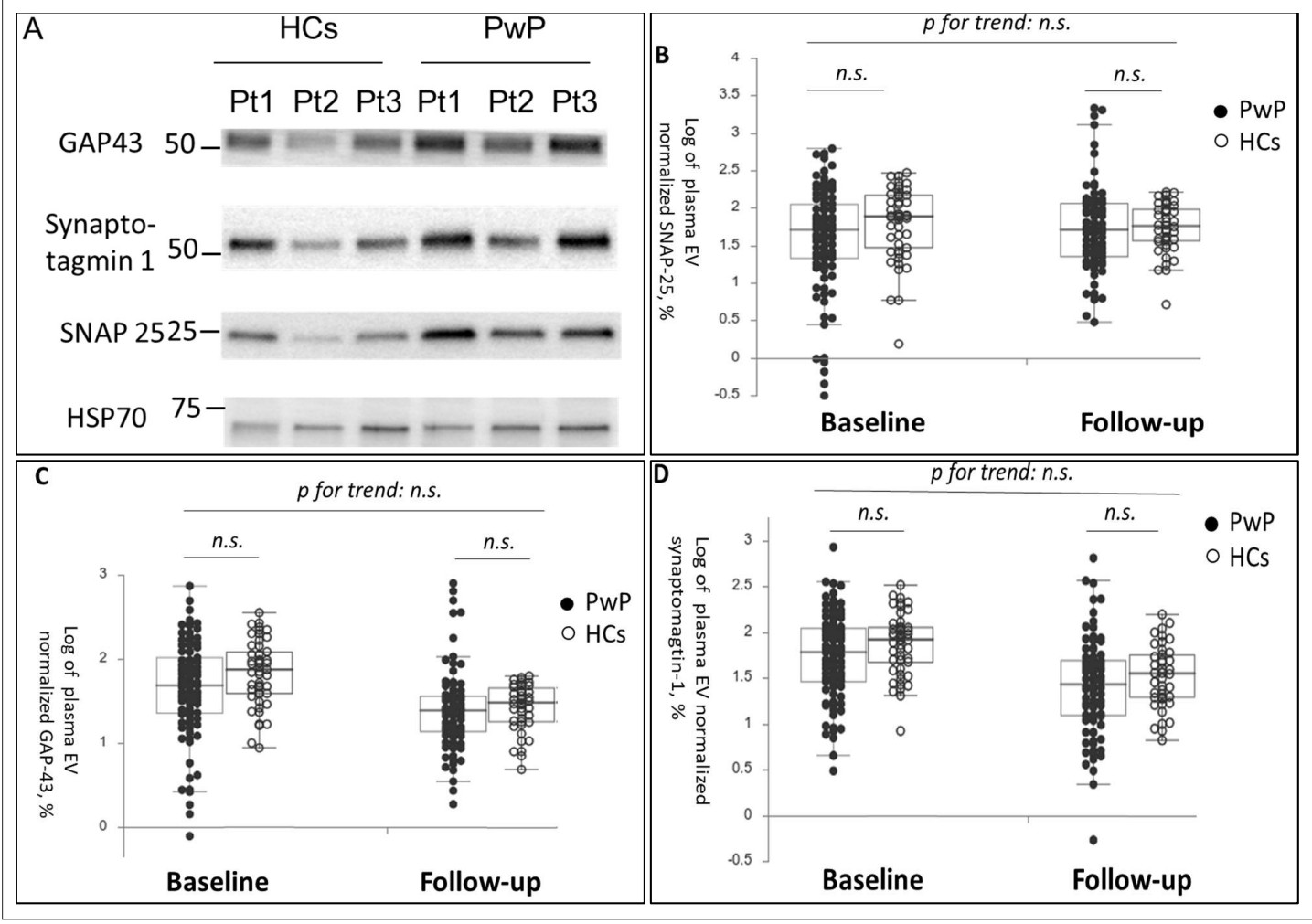

**Figure 1.** Baseline and follow-up synaptic protein levels in plasma extracellular vesicles (EVs) between patients with Parkinson's disease (PwP) and healthy controls (HCs). (**A**) Representative protein blot images of different synaptic proteins, including SNAP-25, GAP-43, and synaptotagmin-1. Heat shock protein 70 (HSP-70) was the protein loading control. (**B–D**) Comparison of plasma SNAP-25, GAP-43, and synaptotagmin-1 levels between PwP and HCs at baseline and follow-up. Data are presented using a dot plot displaying the median and first and third quartile values. n.s., nonsignificant.

The online version of this article includes the following source data for figure 1:

**Source data 1.** The original blot image of the representative GAP-43 (**A**), synaptotagmin-1 (**B**), SNAP-25 (**C**), and HSP-70 (**D**).

EV synaptic proteins exhibited significantly greater deterioration, as assessed using the UPDRS-II scores and PIGD subscores in UPDRS-III. After adjusting for age, sex, and disease duration, repeated-measures analysis of covariance showed that PwP who had elevated levels of any one plasma EV synaptic protein had significantly higher UPDRS-II scores and PIGD subscores at follow-up, but not at baseline, compared to those without elevated levels. This indicates a notably faster deterioration in the patient group with elevated plasma EV synaptic protein levels (**Figure 3**).

## Discussion

Although no significant difference in plasma EV synaptic protein levels was observed between PwP and HCs in the research, alterations in plasma EV synaptic protein levels in PwP were linked to motor decline. In addition, baseline plasma EV synaptic protein levels were associated with clinical outcomes, as assessed using PIGD subscores at follow-up. PwP with elevated baseline levels of any one plasma EV synaptic protein (SNAP-25, GAP-43, or synaptotagmin-1) exhibited significant deterioration in the activities of daily living (as assessed by UPDRS-II scores and PIGD subscores in UPDRS-III) between

**Table 2.** The association between the change in plasma EV synaptic protein abundance (between baseline and follow-up) with the change in clinical severity in motor and cognitive domains (between baseline and follow-up) in people with Parkinson's disease.

A generalized linear model was employed, and the data is presented as coefficient (p-value).

|  | UPDRS-II | UPDRS-III | Tremor | AR | PIGD | MMSE | MoCA |
|---|---|---|---|---|---|---|---|
| SNAP-25 * follow-up | 0.218 (**0.049**) | 0.312 (0.076) | 0.004 (0.432) | 0.016 (0.066) | 0.009 (0.440) | −0.007 (0.932) | 0.048 (0.647) |
| GAP-43 * follow-up | 0.984 (**0.031**) | 1.711 (**0.018**) | 0.001 (0.972) | 0.089 (**0.011**) | 0.073 (0.115) | −0.099 (0.767) | −0.054 (0.901) |
| Synaptomagtin-1 * follow-up | 1.543 (**0.012**) | 2.205 (**0.024**) | 0.007 (0.815) | 0.107 (**0.023**) | 0.109 (0.080) | −0.361 (0.421) | −0.260 (0.661) |

UPDRS = Unified Parkinson's Disease Rating Scale; AR = akinetic rigidity; PIGD = postural instability and gait disturbance; MMSE = Mini-Mental Status Examination; MoCA = Montreal Cognitive Assessment.

baseline and follow-up. The promising potential of plasma EV synaptic proteins as biomarkers for PD, particularly its progression, was indicated.

Significant worsening of UPDRS-II scores and PIGD subscores in UPDRS-III was noted in PwP with higher baseline levels of plasma EV synaptic proteins (first quartile). UPDRS-II is a crucial but underestimated parameter for assessing PwP; it can be used to assess the daily functional capability related to motor symptoms in PwP without temporary drug interruptions, as in the case of UPDRS-III (*Rodriguez-Blazquez et al., 2013*; *Sampaio, 2009*). PIGD subtype is linked to a higher Lewy bodies burden, a rapid decline, and greater risks of cognitive impairment, falls, and mortality compared to the tremor-dominant (TD) subtype. The TD subtype, in contrast, tends to have a relatively benign

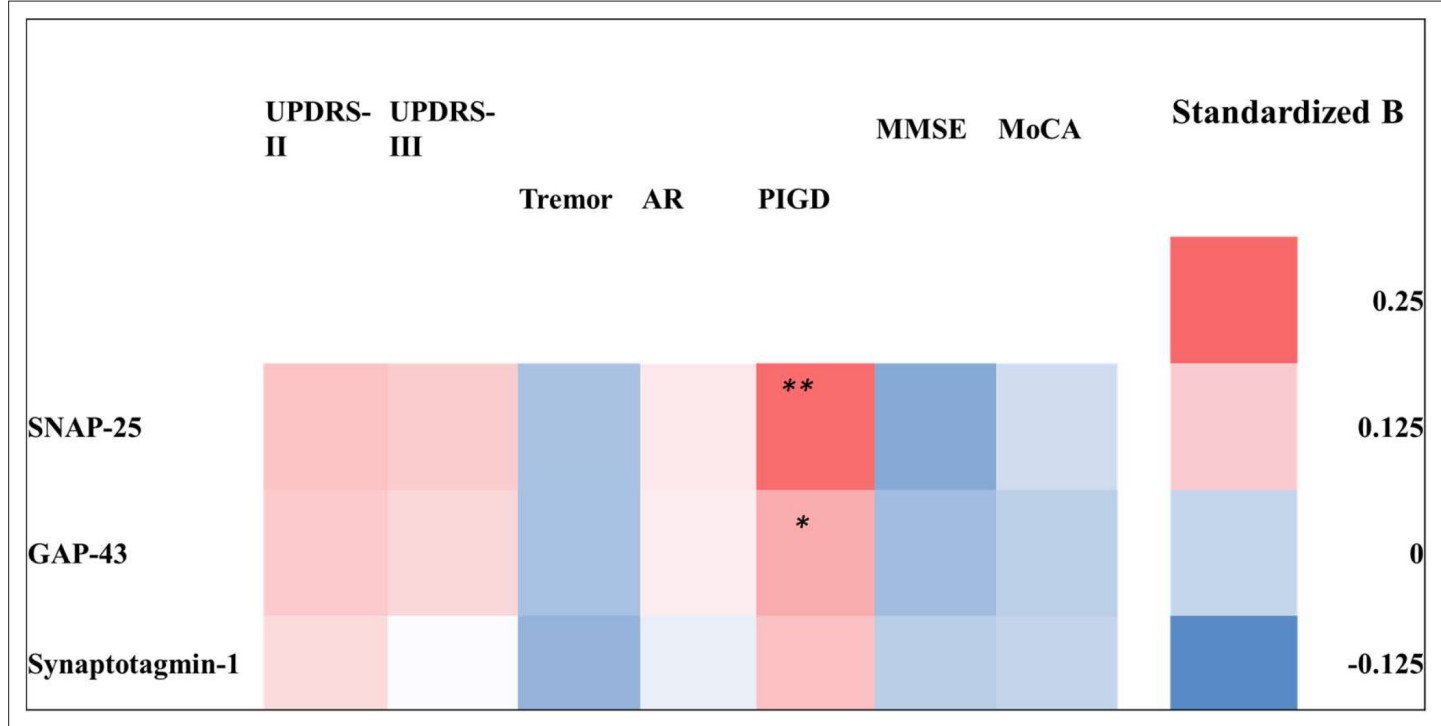

**Figure 2.** Heatmap of the association between baseline plasma extracellular vesicle (EV) synaptic protein levels and clinical assessment parameters at follow-up in patients with Parkinson's disease (PwP). The logistic regression model was used to assess the baseline plasma EV SNAP-25, GAP-43, and synaptotagmin-1 levels. The motor symptoms were assessed based on the Unified Parkinson's Disease Rating Scale (UPDRS)-II and UPDRS-III scores and tremor, akinetic rigidity (AR), and postural instability and gait disturbance (PIGD) subscores, and cognitive function was assessed using the Mini-Mental State Examination (MMSE) and Montreal Cognitive Assessment (MoCA) scores. The association is presented using standardized β values. Detailed results of the regression model are provided in *Supplementary file 1*. *p<0.05; **p<0.01.

**Table 3.** The clinical severity in people with Parkinson's disease with and without elevated (first quartile) baseline plasma extracellular vesicle (EV) synaptosome-associated protein 25 (SNAP-25), growth-associated protein 43 (GAP-43), and synaptotagmin-1. p-Value indicates the inter-group comparisons for the changes.

| Plasma EV | | SNAP-25 | | | GAP-43 | | | Synaptotagmin-1 | | |
|---|---|---|---|---|---|---|---|---|---|---|
| | | L ($n = 74$) | H ($n = 28$) | p | L ($n = 74$) | H ($n = 28$) | p | L ($n = 74$) | H ($n = 28$) | p |
| UPDRS-II | Baseline | 8.26 ± 5.80 | 9.04 ± 4.90 | | 8.53 ± 5.67 | 8.32 ± 5.33 | | 8.46 ± 5.70 | 8.50 ± 5.24 | |
| | Follow-up | 10.38 ± 6.11 | 13.36 ± 6.72 | <0.001 | 10.65 ± 6.30 | 12.64 ± 6.53 | <0.001 | 10.57 ± 6.25 | 12.86 ± 6.58 | <0.001 |
| UPDRS-III | Baseline | 22.31 ± 9.54 | 23.89 ± 8.65 | | 22.22 ± 9.66 | 24.14 ± 8.25 | | 21.97 ± 9.37 | 24.79 ± 8.93 | |
| | Follow-up | 20.01 ± 9.33 | 24.21 ± 8.87 | 0.259 | 19.96 ± 9.25 | 24.36 ± 8.99 | 0.244 | 20.04 ± 9.39 | 24.14 ± 8.70 | 0.145 |
| Tremor | Baseline | 0.34 ± 0.46 | 0.46 ± 0.38 | | 0.34 ± 0.34 | 0.46 ± 0.37 | | 0.33 ± 0.34 | 0.47 ± 0.38 | |
| | Follow-up | 0.27 ± 0.26 | 0.35 ± 0.32 | 0.037 | 0.27 ± 0.26 | 0.35 ± 0.32 | 0.034 | 0.28 ± 0.29 | 0.32 ± 0.22 | 0.014 |
| AR | Baseline | 1.04 ± 0.46 | 1.09 ± 0.44 | | 1.03 ± 0.47 | 1.11 ± 0.42 | | 1.03 ± 0.46 | 1.13 ± 0.45 | |
| | Follow-up | 0.96 ± 0.46 | 1.10 ± 0.42 | 0.300 | 0.95 ± 0.45 | 1.12 ± 0.43 | 0.325 | 0.96 ± 0.46 | 1.10 ± 0.42 | 0.195 |
| PIGD | Baseline | 0.76 ± 0.61 | 0.81 ± 0.44 | | 0.77 ± 0.60 | 0.77 ± 0.45 | | 0.74 ± 0.56 | 0.86 ± 0.58 | |
| | Follow-up | 0.74 ± 0.59 | 1.07 ± 0.75 | 0.023 | 0.78 ± 0.61 | 0.98 ± 0.74 | 0.046 | 0.73 ± 0.56 | 1.11 ± 0.80 | 0.027 |
| MMSE | Baseline | 25.32 ± 4.45 | 25.29 ± 3.29 | | 25.58 ± 4.16 | 24.61 ± 4.11 | | 25.62 ± 3.91 | 24.50 ± 4.69 | |
| | Follow-up | 24.78 ± 5.93 | 25.14 ± 4.64 | 0.483 | 25.05 ± 5.79 | 24.43 ± 5.10 | 0.470 | 25.23 ± 5.61 | 23.96 ± 5.50 | 0.342 |
| MoCA | Baseline | 20.68 ± 6.00 | 21.21 ± 5.00 | | 21.14 ± 5.77 | 20.04 ± 5.62 | | 21.12 ± 5.33 | 20.07 ± 6.69 | |
| | Follow-up | 20.60 ± 6.85 | 21.46 ± 5.37 | 0.834 | 21.10 ± 6.55 | 20.18 ± 6.30 | 0.899 | 21.19 ± 6.33 | 19.93 ± 6.80 | 0.926 |

L = second to fourth quartile at baseline; H = first quartile at baseline; UPDRS = Unified Parkinson's Disease Rating Scale; AR = akinetic rigidity; PIGD = postural instability and gait disturbance; MMSE = Mini-Mental Status Examination; MoCA = Montreal Cognitive Assessment.

course (*Johnson et al., 2016*; *Burn et al., 2006*; *Ren et al., 2020*; *Nutt, 2016*). PwP may convert from the TD subtype to the PIGD subtype during disease progression (*Lee et al., 2019*). Moreover, PIGD-related motor symptoms respond to dopaminergic medications to a lesser extent than do tremor-, akinesia-, and rigidity-related symptoms (*Kempster et al., 2007*), which may more directly reflect the progression of the disease. PwP with elevated levels of any one plasma EV synaptic protein exhibited greater deterioration, as assessed using UPDRS-II scores and PIGD subscores, as observed in the research. High EV synaptic protein levels conventionally indicate increased synaptogenesis; this contrasts the synaptic loss pattern noted in PD. Decreased blood exosomal synaptic protein levels have been reported in patients with AD and frontotemporal dementia (*Goetzl et al., 2016*). However, in PwP who are at the early disease stage, compensatory synaptic sprouting and increased synaptic plasticity are noted in the striatum. This compensation, also known as motor reserve, may temporarily decrease the clinical disease burden. However, the patient would be more vulnerable to deterioration if the compensation is overshadowed by degeneration (*Chung et al., 2020e*). For instance, in one study, PwP who engaged in higher premorbid exercises exhibited milder motor symptoms despite having a similar gradient of striatal dopaminergic reduction at baseline; however, they exhibited more rapid deterioration (*Sunwoo et al., 2017*). The inclusion of primarily early-stage PwP (mean disease duration of less than 3 y) with mild disease burden (baseline UPDRS-II score = 8.49) in the research suggests that elevated plasma EV synaptic protein levels might indicate activation of a compensatory process. Such PwP are at an increased risk of rapid deterioration and disease progression and should be candidates for disease-modifying interventions, including pharmacological and nonpharmacological treatments.

A notable pioneering role is the evaluation of changes in plasma EV synaptic protein levels and elucidating the relationship between these changes and cognitive decline in PwP. Considering the well-established role of synaptic degeneration and plasticity in PD, the theoretical background of the use of these synaptic proteins as plasma EV biomarkers for PD is confirmed. Moreover, most synaptic

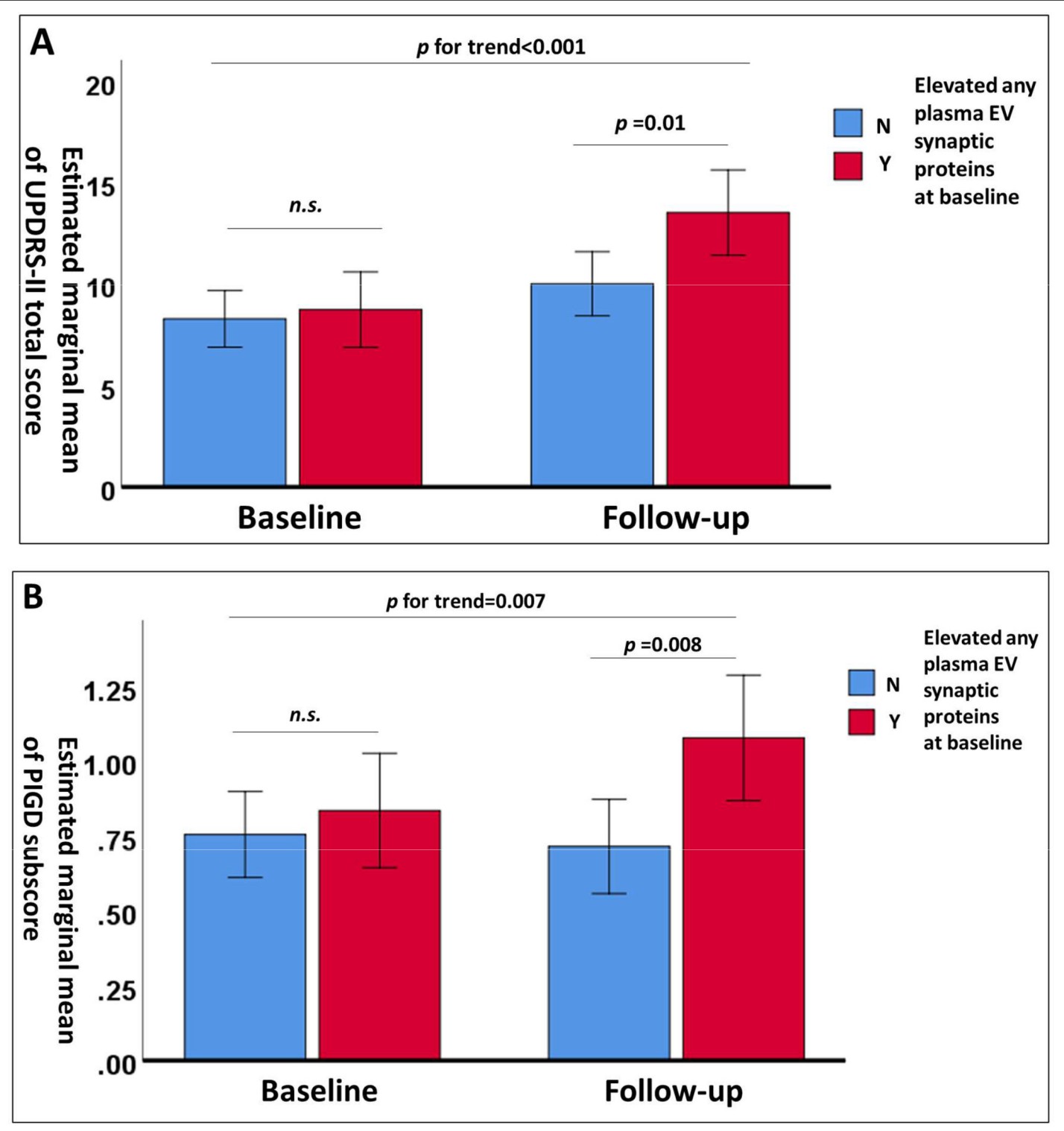

**Figure 3.** Changes in estimated marginal means of Unified Parkinson's Disease Rating Scale (UPDRS)-II total scores. (**A**) and postural instability and gait disturbance (PIGD) subscores (**B**) after adjustment for age, sex, and disease duration in patients with Parkinson's disease (PwP) with (n=36) and without (n=66) elevated levels of any one plasma extracellular vesicle synaptic protein (first quartile) at baseline and follow-up. Data are presented as means with 95% confidence intervals. n.s., nonsignificant.

proteins are neuron derived, thereby preventing contamination from non-neuronal tissues. The significant link between changes in plasma EV synaptic protein levels and changes in UPDRS-II, UPDRS-III, and AR subscore indicates the effectiveness of these proteins in detecting motor decline in PwP. This suggests that these proteins could serve as an objective parameter for future disease-modification clinical trials. Elevated baseline plasma EV synaptic protein levels can also predict rapid deterioration in PwP, highlighting the importance of motor reserve and synaptic plasticity in the progression of PD for future research. Although no single synaptic protein can precisely predict the progression of PD, these protein targets may be considered for inclusion in a biomarker panel. They can be analyzed using an artificial intelligence-assisted artificial neural network, a method commonly used to predict outcomes in various neurological diseases (*Chung et al., 2020b*; *Chung et al., 2020c*; *James et al., 2021*; *Chou et al., 2022*).

There were several limitations to the study. Technically, semiquantitative assessment of plasma EV synaptic protein (SNAP-25, GAP-43, and synaptotagmin-1) levels was performed using western blot analysis. The lack of absolute values, that is, from the results of enzyme-linked immunosorbent assay, limits further clinical application. In addition, because numerous synaptic proteins are involved in the pathogenesis of PD, the three selected proteins may not reflect all features of the synaptic condition. The selection of HSP-70 as the control for the synaptic proteins quantification of plasma EV is not undisputable. Two types of EV proteins are used for this role: membrane proteins and intravesicular proteins. Membrane proteins include CD9, CD63, and CD81; intravesicular proteins include TSG101, annexins, and chaperone proteins, such as HSP-70. Since the expression of HSP-70 is usually steady, the HSP-70 was used as the internal control, but it is a limitation of the present study. Furthermore, the evaluation of the overall plasma EVs rather than specifically focusing on neuron-derived exosomes potentially introduced a bias toward somatic-origin EVs. Nonetheless, it is worth noting that synaptic proteins primarily originate from neurons. Even when considering neuron-derived exosomes, it is important to recognize that they are not exclusively derived from the brain, which can lead to contamination from the peripheral nervous system. The final technical issue in the present study was the relatively small size of the isolated EVs. Despite the primary focus on isolating exosomes, which are the smallest type of EVs, it is important to consider that the presence of small-sized EVs could potentially be attributed to EV fragmentation that occurs during the freezing and thawing processes. Regarding the limitation from the cohort, it is also worth mentioning that the 1-year follow-up period to assess the progression of PD was relatively short and may have been insufficient to detect significant disease progression. On the other hand, the evaluation of motor symptoms took place in a hospital setting. In this context, patients were not asked to stop taking their anti-PD medications. This decision was made due to safety concerns, such as the risk of falls. As a result, specific motor symptoms, particularly tremor and AR, which are more sensitive to medication compared to PIGD, may have been effectively managed by the anti-PD medications. This could potentially explain the improvement in tremor observed between the baseline and 1-year follow-up, especially among PwP with elevated baseline plasma EV synaptic proteins. Additionally, synaptic dysfunction is a frequently observed phenomenon in several neurological diseases, and it is not exclusive to PD. Consequently, the HC group may have included individuals with coexisting neurological conditions in the research, potentially explaining the lack of a significant difference between the PD group and the HCs. However, this approach also illuminates the significance of synaptic dysfunction in the advancement of PD. This insight can be invaluable for monitoring disease progression, particularly in the context of clinical trials focused on disease modification. Choosing the first quartile as a cutoff value of plasma EV synaptic proteins is also one of the limitations of the study. While developing new biomarkers, there was no clear cutoff value as reference for the continuous variable, and percentile is considered for predicting the prognosis (*Duarte, 2021*). Further studies are required to validate this application. Lastly, the results form a mono-centric, small-scale and short-period PD cohort required further validation.

In conclusion, the changes in the levels of plasma EV synaptic proteins, namely SNAP-25, GAP-43, and synaptotagmin-1, are associated with motor decline in PwP. Elevated baseline plasma EV synaptic protein levels can predict increased deterioration of motor function, particularly PIGD symptoms, in PwP. Plasma EV synaptic proteins have the potential to be used as biomarkers of PD progression and detection. A longer longitudinal follow-up is warranted to clearly assess the prognostic efficacy of plasma EV synaptic proteins in PwP.

## Methods

### Study participants

A total of 101 PwP and 43 HCs were enrolled. PD was diagnosed in accordance with the criteria used in another study (*Hughes et al., 1992*). Patients diagnosed as having early-to-mid-stage PD (Hoehn and Yahr stages I–III) were invited to participate in this study. HCs were excluded if they had comorbidities, such as neurodegenerative, psychiatric, or major systemic diseases (malignant neoplasm or chronic kidney disease). HCs were mainly recruited from neurological outpatient clinics; they had minor chronic health conditions, such as hypertension, diabetes, or hyperlipidemia. The research protocol was approved by the Joint Institutional Review Board of Taipei Medical University (approval nos. N201609017 and N201801043).

### Clinical assessments

The participants' background data were obtained through a personal interview. Their cognitive function was assessed by trained nurses using the Taiwanese versions of the MMSE and MoCA. The severity of PD was assessed using parts I–III of the UPDRS during clinic visits. PwP were assumed to be in their 'on' time. Tremor, AR, and PIGD subscores were calculated from the subitems in UPDRS-III as described previously (*Lewis et al., 2005*), with some modifications.

### Plasma EV isolation and characterization

Venous blood was collected by from PwP and HCs after their outpatient clinic (non-fasting) by 21-gauge needle, and the plasma was isolated through centrifugation at 13,000 × $g$ for 20 min immediately. Plasma was stored in the −80°C freezer before EV isolation. Plasma EVs were isolated from 1 mL of plasma using exoEasy Maxi Kit (QIAGEN, Valencia, CA) in accordance with the manufacturer's instructions and stored in the −80°C freezer. It was a membrane-based affinity binding step to isolate exosomes and other EVs without relying on a particular epitope. The isolated plasma EVs were then eluted and stored. Usually, 400 µL of eluate is obtained per mL of plasma. The isolated plasma EVs underwent validation in accordance with the guidelines of the International Society of Extracellular Vesicles. This validation process encompassed several steps. First, the presence of markers was confirmed, including CD63 (ab59479, Abcam, Cambridge, UK), CD9 (ab92726, Abcam), and tumor susceptibility gene 101 protein (GTX118736, GeneTex, CA), along with the absence of cytochrome *c* (ab110325; Abcam). Second, physical characterization was performed using nanoparticle tracking analysis. This analysis revealed that the majority of EVs were primarily within the 50–100 nm size range. Third, the morphology of the EVs was examined through electron microscopy analysis. The validation had been described previously (*Chung et al., 2020d*; *Chung et al., 2020a*; *Chung et al., 2021a*).

### Quantification of plasma EV synaptic proteins

The isolated plasma EVs were directly lysed using protein sample buffer (RIPA Lysis Buffer, Millipore) and analyzed using protein sodium dodecyl sulfate–polyacrylamide gel electrophoresis. Antibodies against SNAP-25 (GeneTex, GTX113839, 1:1000), GAP-43 (GeneTex, GTX114124, 1:5000), and synaptotagmin-1 (GeneTex, GTX127934, 1:1000) were used for the analysis. The antibodies were prepared in Tris-buffered saline containing 0.1% Tween 20 and 5% bovine serum albumin. Secondary antibodies, including antimouse immunoglobulin G (IgG)-conjugated horseradish peroxidase (HRP; 115-035-003) and antirabbit IgG-conjugated HRP (111-035-003), were purchased from Jackson ImmunoResearch. Protein blot intensities were quantified using ImageJ software. The expression levels of plasma EV synaptic proteins were normalized to that of heat shock protein 70 (Proteintech, Cat# 10995-1-AP, 1:2000). For each participant, equal volume of EV suspension (5 µL) was applied to the protein quantification. To ensure that the data could be compared between different gels, all the data were normalized to the average of the control group in the same gel.

### Statistical analyses

All statistical analyses were performed using SPSS for Windows 10 (version 26; SPSS Inc, Chicago, IL). A linear mixed model was used to assess whether the changes in plasma EV synaptic protein levels differed between PwP and HCs after adjustment for age and sex. A generalized linear model was used to determine the association between the changes in plasma EV synaptic protein levels and the

changes in clinical parameters in PwP after adjustment for age, sex, and disease duration. Multivariate logistic regression was performed to assess the association between plasma EV synaptic proteins and clinical parameters at follow-up in PwP after adjustment for age, sex, and disease duration. Repeated-measures analysis of covariance with estimated marginal means was employed to compare the changes in clinical parameters between baseline and follow-up in PwP with elevated baseline levels (first quartile) of any one plasma EV synaptic protein. Finally, p-values<0.05 were considered statistically significant.

## Acknowledgements

The Ministry of Science and Technology, Taiwan, funded this study (MOST 110-2314-B-038-096 and NSC 111-2314-B-038-136).

## Additional information

### Funding

| Funder | Grant reference number | Author |
|---|---|---|
| Minister of Science and Technology | MOST 110-2314-B-038-096 | Chien-Tai Hong |
| National Science Council | 111-2314-B-038-136 | Chien-Tai Hong |

The funders had no role in study design, data collection and interpretation, or the decision to submit the work for publication.

### Author contributions

Chien-Tai Hong, Conceptualization, Resources, Data curation, Formal analysis, Funding acquisition, Investigation, Methodology, Writing – original draft, Project administration; Chen-Chih Chung, Resources, Data curation, Software, Funding acquisition, Investigation, Methodology, Writing – review and editing; Ruan-Ching Yu, Data curation, Formal analysis, Methodology; Lung Chan, Conceptualization, Resources, Data curation, Software, Validation, Investigation, Writing – review and editing

### Author ORCIDs

Chien-Tai Hong https://orcid.org/0000-0002-7448-1041
Lung Chan https://orcid.org/0000-0001-5795-4460

### Ethics

Human subjects: This study was approved by the Joint Institutional Review Board of Taipei Medical University (TMU-JIRB approval no. N201609017 and N201801043). Written informed consent was obtained from all participants for participation in the study.

Reviewer #1 (Public Review): https://doi.org/10.7554/eLife.87501.3.sa1
Reviewer #2 (Public Review): https://doi.org/10.7554/eLife.87501.3.sa2
Author Response https://doi.org/10.7554/eLife.87501.3.sa3

## Additional files

### Supplementary files

• Supplementary file 1. Association between the baseline plasma EV synaptic proteins with the clinical severity in people with Parkinson's disease at follow-up with the adjustment of age, sex, disease duration, and the baseline severity of corresponding item, presented as standardized β and p-values. UPDRS, Unified Parkinson's Disease Rating Scale; AR, akinetic rigidity; PIGD, postural instability and gait disturbance; MMSE, Mini-Mental Status Examination; MoCA, Montreal Cognitive Assessment.

• MDAR checklist

## Data availability

The dataset is deposited in Dryad: https://doi.org/10.5061/dryad.4qrfj6qh9.

The following dataset was generated:

| Author(s) | Year | Dataset title | Dataset URL | Database and Identifier |
|---|---|---|---|---|
| Hong CT, Chung CC, Yu RC, Chan L | 2024 | Plasma extracellular vesicle synaptic proteins as biomarkers of clinical progression in patients with Parkinson's disease | https://doi.org/10.5061/dryad.4qrfj6qh9 | Dryad Digital Repository, 10.5061/dryad.4qrfj6qh9 |

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
