## [Editor Report · eLife assessment]

This **useful** study presents data regarding the presence of synaptic proteins in the extracellular vesicle pool present in the blood of Parkinson's patients and non-Parkinson neurological outpatients, trying to correlate changes in such levels with the progression of Parkinson's symptoms. The results are semi-quantitative and preliminary, suggesting that these biomarkers could be used in the follow-up of a specific group of Parkinson’s patients. The evidence is **incomplete** at this point, and more quantitative approaches are required to propose this correlation. The isolation of extracellular vesicles was appropriate as revealed by their sizes, but they are not exclusively from neuronal origin. The presented approach is not ready to be used in the clinical setting.

---

## [Referee Report · Reviewer #1 (Public Review)]

The study isolated extracellular vesicles (EV) from healthy controls (HCs) and Parkinson patients (PwP), using plasma from the venous blood of non-fasting people. Such EVs were characterized and validated by the presence of markers, their size, and their morphology. The main aim of the manuscript is to correlate the presence of synaptic proteins, namely SNAP-25, GAP-43, and SYNAPTOTAGMIN-1, normalized with HSP70, with the clinical progression of PwP. Changes in synaptic proteins have been documented in the CSF of Alzheimer's and Parkinson's patients. The demographics of participants are adequately presented. One important limiting, as well as puzzling aspect, is the fact that authors did not find differences between groups at the beginning of the study nor after one year, after age and sex adjustment.

---

## [Referee Report · Reviewer #2 (Public Review)]

Hong and collaborators investigated variations in the amount of synaptic proteins in plasma extracellular vesicles (EV) in Parkinson's Disease (PD) patients on one-year follow-up. Their findings suggest that plasma EV synaptic proteins may be used as clinical biomarkers of PD progression.

It is a preliminary study using semi-quantitative analysis of synaptic proteins.

The authors have a cohort of PD patients with clinical examination and a know-how on EV purification. Regarding this latter part, they may improve their description of EV purification. EV may be broken into smaller size EV after freezing. Does it explain the relatively small size in their EV preparation? Do the authors refer to the MISEV guidelines for EV purity? Regarding synaptic protein quantification, the choice of western blotting may not be the best one. ELISA and other multiplex arrays are available. How the authors do justify their choice? Do the authors try to sort plasma EV by membrane-associated neuronal EV markers using either vesicle sorting or immunoprecipitation?

Many technical aspects may be improved. Such technical questions weakened the authors' conclusions.

---

## [Author Response]

The following is the authors’ response to the original reviews.

**Reviewer #1 (Public Review):**
The study isolated extracellular vesicles (EV) from healthy controls (HCs) and Parkinson patients (PwP), using plasma from the venous blood of non-fasting people. Such EVs were characterized and validated by the presence of markers, their size, and their morphology. The main aim of the manuscript is to correlate the presence of synaptic proteins, namely SNAP-25, GAP-43, and SYNAPTOTAGMIN-1, normalized with HSP70, with the clinical progression of PwP. Changes in synaptic proteins have been documented in the CSF of Alzheimer's and Parkinson's patients. The demographics of participants are adequately presented.One important limiting, as well as puzzling aspect, is the fact that authors did not find differences between groups at the beginning of the study nor after one year, after age and sex adjustment.

Response: Thanks for your comments. We acknowledge your observation that the absence of a discernible difference in plasma EV synaptic protein levels between the PD and control subjects constitutes a significant limitation of our study. This outcome could be attributed to the fact that the controls were recruited from the neurology outpatient clinic, representing a group that could be considered "sub-healthy." Moreover, these individuals are not exempt from aging-related neurodegenerative processes. Considering that our PD subjects are in the early stages of the disease (with a mean disease duration of less than 3 years) and that synaptic dysfunction is a broader indicator rather than specific to PD, these factors could collectively contribute to the lack of distinction between the PD and control groups.

However, our primary intention was also to explore the potential of plasma EV synaptic proteins as predictive markers for disease progression in PD. In this regard, we have identified their applicability within the current PD cohort. We are committed to conducting further follow-up with these study subjects over an extended duration to delve deeper into these findings.

We revised the following statement in the discussion part to address this issue as following “Additionally, synaptic dysfunction is a frequently observed phenomenon in several neurological diseases, and it is not exclusive to PD. Consequently, the HC group in our current study may have included individuals with coexisting neurological conditions, potentially explaining the lack of a significant difference between the PD group and the HCs. However, this approach also illuminates the significance of synaptic dysfunction in the advancement of PD. This insight can be invaluable for monitoring disease progression, particularly in the context of clinical trials focused on disease modification.”

Tables in general are hard to follow. Specifically, Table 2 does not convey a clear message nor in the text of the Table itself, and the per 100% of change needs to be explained in the corresponding legend.

Response: Thanks for your comment. In Table 2, our aim was to demonstrate the association between the change of plasma EV synaptic proteins with the change of clinical severity, and presented as coefficient (p value). We apologize for any prior ambiguity in the main text's description of these results and have since made revisions to enhance clarity.

Regarding the "per 100% change," this is due to the quantification of plasma EV synaptic proteins being based on a semi-quantitative Western blot method. Each measurement was normalized by the average baseline plasma synaptic protein levels of healthy controls (HCs). The term "per 100% change" denotes the increase or decrease in plasma EV synaptic protein abundance relative to the average baseline levels observed in healthy controls. We apologize for any confusion caused and removed this term. In addition, we rephrased the statement to ensure better understanding and readability in the Table legend of revised manuscript as following “The association between the change of plasma EV synaptic proteins abundance (between baseline and follow-up) with the change of clinical severity in motor and cognitive domains (between baseline and follow-up) in people with Parkinson’s disease. A generalized linear model was employed and the data was presented as coefficient (p value).”

It is only when PwP were classified as a first quartile that a significantly greater deterioration was found. However, in the case of tremor, the top 25% had values going from 0.46-0.47 to 0.32-0.35, whereas the lower three quarters went from 0.33-0.34 to 0.27-0.28 depending on the protein analyzed. This needs to be clarified in the text.

Response: Thanks for your comments. As per the unified Parkinson's disease rating score (UPDRS), a higher score indicates greater severity of symptoms. Regarding tremor, we observed a general trend of improvement in both groups. PwP with elevated baseline plasma EV proteins had a trendy of worse tremor score at baseline, and the improvement was significantly better than the rest of PwP. This improvement seems to contradict the progressive nature of PD, and one possible explanation could be the alleviation of symptoms due to medication usage. The assessment of motor symptoms took place within the hospital setting, where we refrained from requesting patients to withhold their anti-PD medications due to concerns about safety issues such as falls. Consequently, certain motor symptoms might have been effectively controlled by the anti-PD medication. Traditionally, symptoms like tremor and rigidity (as reflected by the akinetic rigidity score) respond well to medications, while postural instability and gait disturbance (PIGD) are less responsive. In our cohort, we noted an improvement in tremor scores and stability in akinetic rigidity (AR) scores. Conversely, PD patients with higher baseline plasma EV synaptic protein levels exhibited notable progression in PIGD scores. These findings have been documented in the results section and discussed comprehensively within the revised manuscript as following “On the other hand, the evaluation of motor symptoms occurred in a hospital setting where we did not ask patients to stop taking their anti- PD medications due to safety concerns like the risk of falls. As a result, specific motor symptoms, particularly tremor and AR, which are more sensitive to medication compared to PIGD, may have been effectively managed by the anti-PD medications. This could potentially explain the improvement in tremor observed between the baseline and one-year follow-up, especially among PwP with elevated baseline plasma EV synaptic proteins.”

Table 3 is hard to read and some of the values seem repetitive, especially for tremor, AR, and PIGD. It looks as if Figure 2 represents the same information as Table 3.

Response: Thanks for your information. We have ensured the accuracy of the results presented in Table 2. While some of the entries may appear similar, they do indeed possess distinct differences.

To enhance readability, we streamlined the information in Table 3 by removing the p-values from the intra-group comparisons between baseline and the 1-year follow-up within each domain. We retained the original p-values for trend related to the inter-group comparisons for changes. Detailed information has been relocated to the supplementary section of the revised manuscript. In Figure 2, we illustrated the relationship between baseline plasma extracellular vesicle (EV) synaptic protein levels and the clinical assessment parameters during follow-up in patients with Parkinson's disease (PwP). This portrayal is distinct from the information depicted in Table 3.

If you had concerns about the resemblance between Table 3 and Figure 3, please note that the values in Table 3 represent raw scores, while the values in Figure 3, namely the estimated marginal means, are the "adjusted" scores for UPDRS-II and PIGD at baseline and follow-up. These adjustments encompass age, sex, and disease duration. We sincerely apologize for any lack of clarity in our previous description and have since revised it accordingly.

The text and figure legends are not helpful in guiding the reader to understand the presented information.

Response: Thanks for your comments and we apologized for the unclear statement. We revised the figure legend and the main text for better understanding of the readers.

**Reviewer #2 (Public Review):**
Hong and collaborators investigated variations in the amount of synaptic proteins in plasma extracellular vesicles (EV) in Parkinson's Disease (PD) patients on one-year follow-up. Their findings suggest that plasma EV synaptic proteins may be used as clinical biomarkers of PD progression.It is a preliminary study using semi-quantitative analysis of synaptic proteins.

Response: Thanks for your comments. The present study represents the initial phase of our investigation into the role of plasma EV synaptic proteins within our PD cohort. Our findings have revealed the potential predictive significance of these synaptic proteins in relation to PD progression. We are committed to conducting further follow-up with these study subjects over an extended period.

Furthermore, it's important to acknowledge that the semi-quantitative approach employed to assess protein abundance was a limitation of this study. This limitation stems from the low concentration of plasma EV synaptic proteins, which restricts the feasibility of utilizing techniques such as ELISA or other quantitative methods for protein assessment. We have duly acknowledged this limitation within the scope of the present study as following “Semiquantitative assessment of plasma EV synaptic protein (SNAP-25, GAP-43, and synaptotagmin-1) levels was performed using western blot analysis. The lack of absolute values limits further clinical application.”

Moving forward, we intend to adopt alternative EV isolation methods that enable the extraction of a larger abundance of plasma EV proteins, facilitating more accurate quantitative assessments. In addition, a longer longitudinal follow-up is warranted to clearly assess the prognostic efficacy of plasma EV synaptic proteins in PwP, which we had mentioned in the manuscript.

The authors have a cohort of PD patients with clinical examination and a know-how on EV purification. Regarding this latter part, they may improve their description of EV purification. EV may be broken into smaller size EV after freezing. Does it explain the relatively small size in their EV preparation? Do the authors refer to the MISEV guidelines for EV purity?

Response: Thanks for your comments. In the previous manuscript, we provided a relatively detailed account of the procedures related to EV isolation and validation (https://doi.org/10.1096/fj.202100787R). In the revised manuscript, we added some information about the principle of the EV isolation kit, and the validation antibody as following “Plasma EVs were isolated from 1 mL of plasma by exoEasy Maxi Kit (Qiagen, Valencia, CA, USA), a membrane-based affinity binding step to isolate exosomes and other EVs without relying on a particular epitope, in accordance with the manufacturer’s instructions and storaged in the −80。C freezer. The isolated plasma EVs were then eluted and stored. Usually, 400 μL of eluate is obtained per mL of plasma. The isolated plasma EVs were validated according to the International Society of Extracellular Vesicles guidelines, which include1.markers, including the presence of CD63 (ab59479, Abcam, Cambridge, UK), CD9(ab92726, Abcam, Cambridge, UK), tumor susceptibility gene 101 protein (GTX118736, GeneTex, CA, USA) and negative of cytochrome c (ab110325; Abcam, Cambridge, UK) 2. Physical characterization through the nanoparticle tracking analysis, which demonstrated the majority of the size of EV are mainly within 50-100nm 3. The morphology from the electron microscopy analysis. The validation had been described previously [29-31]. “

It's important to note that our primary focus was on exosomes, the smallest subtype of EVs. Through nanoparticle tracking analysis, we observed that the majority of isolated EVs fell within the diameter range of 50-150nm, exhibiting significant surface marker (i.e. CD63 and CD9) expression. Moreover, electron microscopy confirmed their vesicular morphology. These meticulously validated EVs were promptly analysed post-isolation.

However, we acknowledge that the plasma obtained from study participants might have undergone freezing prior to EV isolation. This freezing process has the potential to diminish the yield rate of EVs and result in some degree of fragmentation. We have duly included this issue as a limitation in our revised manuscript as following “The final technical issue in the present study was the relatively small size of the isolated EVs. Despite the primary focus on isolating exosomes, which are the smallest type of EVs, it's important to consider that the presence of small-sized EVs could potentially be attributed to EV fragmentation that occurs during the freezing and thawing processes.”

Regarding synaptic protein quantification, the choice of western blotting may not be the best one. ELISA and other multiplex arrays are available. How the authors do justify their choice?

Response: Thanks for your comments. We appreciate your input regarding the semi-quantitative western blot analysis not being the most optimal approach. Owing to the limited quantity of isolated plasma EVs and the significant protein abundance of synaptic proteins within these EVs, we did explore the use of an ELISA assay. However, it's worth noting that for a specific subset of the samples, the readout obtained was lower than the lower limit of detection of the ELISA kit. In response, we have incorporated this point as limitation within the discussion section of the revised manuscript as following “Semiquantitative assessment of plasma EV synaptic protein (SNAP-25, GAP-43, and synaptotagmin-1) levels was performed using western blot analysis. The lack of absolute values, i.e. from the results of enzyme-linked immunosorbent assay, limits further clinical application.”

Do the authors try to sort plasma EV by membrane-associated neuronal EV markers using either vesicle sorting or immunoprecipitation?

Response: Thanks for your comments. The current study did not specifically isolate neuron-derived extracellular vesicles (EVs), potentially introducing some bias to the results. However, it's important to note that synaptic proteins, such as SNAP-25, exhibit a high degree of neuron-specific expression, with a predominant presence in the brain (as indicated by https://www.proteinatlas.org/ENSG00000132639-SNAP25/tissue). Given this context, the limitation of not analyzing neuron-derived EVs could be mitigated to some extent. In response, we have incorporated this point as limitation within the discussion section of the revised manuscript as following “Furthermore, this study evaluated the overall plasma EVs rather than specifically focusing on neuron-derived exosomes, potentially introducing a bias towards somatic-origin EVs. Nonetheless, it is worth noting that synaptic proteins primarily originate from neurons. Even when considering neuron-derived exosomes, it's important to recognize that they are not exclusively derived from the brain, which can lead to contamination from the peripheral nervous system.”

Many technical aspects may be improved. Such technical questions weakened the authors' conclusions.

Response: Thanks for your comments. We recognize that the aforementioned issues represent limitations of our current study. In response, we have incorporated these points as limitations, including the semi-quantitative assessments, the isolation of total but not neuron-derived exosomes in the plasma, and the short follow-up time within the discussion section of the revised manuscript.

The discussion is pretty long to justify the data. It may be shortened by adding some information in the introduction.

Response: Thanks for your comments. We have repositioned a statement from the second paragraph of the discussion to the introduction. This adjustment serves to enrich the background understanding of the link between synaptic dysfunction and neurodegenerative diseases.